# *Adaptive Chameleon or Stubborn Sloth:* Revealing the Behavior of Large Language Models in Knowledge Conflicts

**Jian Xie**[♠][*]    **Kai Zhang**[♣][*]    **Jiangjie Chen**[♠]    **Renze Lou**[♡]    **Yu Su**[♣]

[♠]School of Computer Science, Fudan University
[♣]The Ohio State University [♡]The Pennsylvania State University
jianxie22@m.fudan.edu.cn, {zhang.13253, su.809}@osu.edu

## Abstract

By providing external information to large language models (LLMs), tool augmentation (including retrieval augmentation) has emerged as a promising solution for addressing the limitations of LLMs' static parametric memory. However, how receptive are LLMs to such external evidence, especially when the evidence conflicts with their parametric memory? We present the first comprehensive and controlled investigation into the behavior of LLMs when encountering knowledge conflicts. We propose a systematic framework to elicit high-quality parametric memory from LLMs and construct the corresponding counter-memory, which enables us to conduct a series of controlled experiments. Our investigation reveals seemingly contradicting behaviors of LLMs. On the one hand, different from prior wisdom, we find that LLMs can be *highly receptive* to external evidence even when that conflicts with their parametric memory, given that the external evidence is coherent and convincing. On the other hand, LLMs also demonstrate a strong *confirmation bias* when the external evidence contains some information that is consistent with their parametric memory, despite being presented with conflicting evidence at the same time. These results pose important implications that are worth careful consideration for the further development and deployment of tool- and retrieval-augmented LLMs. Resources are available at https://github.com/OSU-NLP-Group/LLM-Knowledge-Conflict.

## 1 Introduction

After pre-training on massive corpora, large language models (LLMs) (Brown et al., 2020; Chowdhery et al., 2022; Ouyang et al., 2022; OpenAI, 2022; 2023; Zeng et al., 2023; Touvron et al., 2023a) have formed a wealth of **parametric memory**, such as commonsense and factual knowledge (Petroni et al., 2019; Li et al., 2022; Zhao et al., 2023). However, such parametric memory may be inaccurate or become outdated (Liska et al., 2022; Luu et al., 2022) due to misinformation in the pre-training corpus or the static nature of parametric memory, known to be a major cause for hallucinations (Elazar et al., 2021; Shuster et al., 2021; Ji et al., 2023).

Tool[1] (Schick et al., 2023; Qin et al., 2023) or retrieval augmentation (Mallen et al., 2022; Shi et al., 2023b; Ram et al., 2023) has emerged as a promising solution by providing external information as new evidence to LLMs, such as ChatGPT Plugins and New Bing. However, external evidence, inevitably, could conflict with LLMs' parametric memory. We refer to external evidence that conflicts with parametric memory as **counter-memory**. In this paper, we seek to answer the question: *how receptive are LLMs to external evidence, especially counter-memory?* A solid understanding of this question is an essential stepping stone for wider application of tool-augmented LLMs. Not only does this relate to overcoming the limitations of LLM's static parametric memory, but it is also associated

---

[*]The first two authors contributed equally. Work done during Jian Xie's internship at OSU NLP Group.
[1]In the rest of the paper we use "tool-augmented LLMs" because retrievers are one type of tools, but tools are not limited to retrievers (consider, e.g., a question answering tool).

with direct safety concerns. For example, what if a third-party tool, either by the developer or hijacked by attackers, intentionally returns disinformation? Will LLMs be deceived?

We present the first comprehensive and controlled investigation into the behavior of LLMs when encountering counter-memory. A key challenge lies in how to construct the counter-memory. Prior work employs various heuristics, such as negation injection (Niu & Bansal, 2018; Kassner et al., 2021; Gubelmann & Handschuh, 2022) and entity substitution (Longpre et al., 2021; Zhou et al., 2023), and finds that language models (both large and small) tend to be stubborn and cling to their parametric memory. However, such heuristic word-level editing results in incoherent counter-memory (see an example in Section 4.1), which may make it trivial for LLMs to detect and thus neglect the constructed counter-memory. It is unclear how the prior conclusions translate to real-world scenarios, where counter-memory is more coherent and convincing.

We propose a systematic framework to elicit the parametric memory of LLMs and construct the corresponding counter-memory. We design a series of checks, such as entailment from parametric memory to the answer, to ensure that the elicited parametric memory is indeed the LLM's internal belief. For the counter-memory, instead of heuristically *editing* the parametric memory, we instruct an LLM to directly *generate* a coherent passage that factually conflicts with the parametric memory. After obtaining a large pool of parametric memory and counter-memory pairs, we then examine LLMs' behavior in different knowledge conflict scenarios, including 1) when only counter-memory is present as external evidence and 2) when both parametric memory and counter-memory are present.

Our investigation leads to a series of interesting new findings. We highlight the following:

- *LLMs are highly receptive to external evidence* if that is the only evidence, even when it conflicts with their parametric memory. This contradicts the prior wisdom (Longpre et al., 2021), and we attribute this to the more coherent and convincing counter-memory constructed through our framework. On the other hand, this also suggests that *LLMs may be easily deceived* by, e.g., disinformation from malicious (third-party) tools.
- However, with *both* supportive and contradictory evidence to their parametric memory, LLMs show a strong *confirmation bias* (Nickerson, 1998) and tend to cling to their parametric memory. This reveals a potential challenge for LLMs to unbiasedly orchestrate multiple pieces of conflicting evidence, a common situation encountered by generative search engines.

## 2  RELATED WORK

**Parametric Memory in Language Models**    After pre-training, language models have internalized a vast amount of knowledge into their parameters (Roberts et al., 2020; Jiang et al., 2020), also known as parametric memory. Many past studies have explored the elicitation of parametric memory in language models, such as commonsense or factual knowledge probing (Petroni et al., 2019; Lin et al., 2020; Zhang et al., 2021; West et al., 2022; Chen et al., 2023; Wang et al., 2023). Such parametric memory could help solve downstream tasks (Wang et al., 2021; Yu et al., 2023; Sun et al., 2023). However, previous work has discovered that language models only memorize a small portion of the knowledge they have been exposed to during pre-training (Carlini et al., 2021; 2023) due to model's limited memorization abilities. In addition, the parametric memory may become outdated (Lazaridou et al., 2021; De Cao et al., 2021). Such incorrect and outdated parametric memory may show as hallucinations (Elazar et al., 2021; Shuster et al., 2021; Ji et al., 2023). Although some methods are proposed to edit knowledge in language models (Dai et al., 2022; Meng et al., 2022; 2023), they typically require additional modifications on model weights without evaluating the consequences on models' other aspects such as performances and are limited to factual knowledge.

**Tool-augmented Language Models**    To address the limitations of parametric memory, external tools such as retrievers are used to augment language models with up-to-date information, namely tool-augmented (Nakano et al., 2021; Yao et al., 2023; Qin et al., 2023; Schick et al., 2023; Lu et al., 2023) or retrieval-augmented (Guu et al., 2020; Khandelwal et al., 2020; Izacard & Grave, 2021; Borgeaud et al., 2022; Zhong et al., 2022) language models. Such a framework, which has proven its efficacy in enhancing large language models (Shi et al., 2023b; Ram et al., 2023; Mallen et al., 2022), is adopted in real-world applications such as New Bing and ChatGPT Plugins. Inevitably, the external evidence could conflict with the parametric memory. However, the behavior of LLMs in

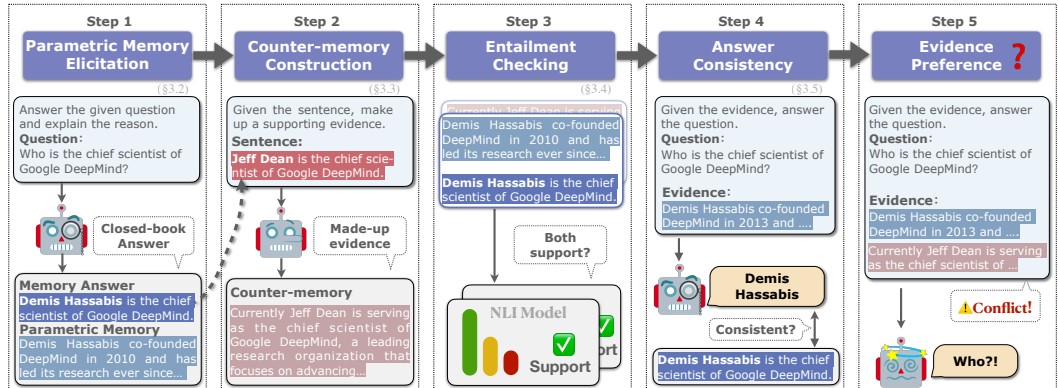

Figure 1: Our framework for simulating knowledge conflicts. In Step 1, we elicit LLMs' memory answer and parametric memory in a closed-book QA fashion. In Step 2, we construct counter-answer to memory answer with heuristic rules, for which ChatGPT generates supporting counter-memory with instructions. To uphold evidence quality, we conduct entailment checking (Step 3) and answer consistency (Step 4) to filter unqualified examples. All experiments are implemented under zero-shot setting to avoid the bias introduced by demonstrations.

knowledge conflict scenarios remains under-explored, and unraveling it holds significance for wider applications of tool-augmented LLMs.

**Knowledge Conflict** To perform controlled experiments, knowledge conflict is often simulated with counter-memory constructed upon parametric memory. Heuristic counter-memory construction methods such as negation injection (Niu & Bansal, 2018; Kassner et al., 2021; Petroni et al., 2020; Pan et al., 2021) have been developed. Furthermore, entity substitution (Longpre et al., 2021; Chen et al., 2022; Si et al., 2023; Zhou et al., 2023) replaces all mentions of the answer entity in parametric memory with other entities to construct counter-memory. However, these methods are limited to word-level editing, leading to low overall coherence in the counter-memory. We instead instruct LLMs to generate counter-memory from scratch to ensure high coherence.

## 3 EXPERIMENTAL SETUP

In this section, we describe our framework for eliciting high-quality parametric memory from LLMs and constructing the corresponding counter-memory, as well as the evaluation metrics.

### 3.1 DATASETS

Following prior work (Longpre et al., 2021; Chen et al., 2022), we adopt question answering (QA) task as the testbed for knowledge conflict experiments. In addition to an entity-based QA dataset (POPQA), we include a multi-step reasoning dataset (STRATEGYQA) for diversifying the questions studied in the experiments. Specifically,

- **POPQA** (Mallen et al., 2022) is an entity-centric QA dataset that contains 14K questions. Data for POPQA originates from triples in Wikidata. Employing custom templates tailored to relationship types, the authors construct questions through the substitution of the subject within knowledge triples. POPQA defines the *popularity* of a question based on the monthly Wikipedia page views associated with the entity mentioned in the question.
- **STRATEGYQA** (Geva et al., 2021) is a multi-step fact reasoning benchmark that necessitates the implicit question decomposition into reasoning steps. The questions are built around Wikipedia terms and cover a wide range of strategies, which demand the model's capability to select and integrate relevant knowledge effectively. The language model is expected to provide a True or False answer.

Table 1: The correctness of LLMs responses in closed-book QA fashion (Step 1 in Figure 1). We examine eight LLMs, including three closed-source LLMs and five open-source LLMs.

| Models | POPQA | | | STRATEGYQA | | |
|---|---|---|---|---|---|---|
| | Correct | Wrong | Unknown | Correct | Wrong | Unknown |
| *Closed-source LLMs* | | | | | | |
| ChatGPT (OpenAI, 2022) | 44.6 | 44.4 | 11.0 | 67.4 | 30.7 | 1.9 |
| GPT-4 (OpenAI, 2023) | 50.8 | 48.7 | 0.5 | 77.3 | 22.7 | 0.0 |
| PaLM2 (Anil et al., 2023) | 32.9 | 67.1 | 0.0 | 67.9 | 32.1 | 0.0 |
| *Open-source LLMs* | | | | | | |
| Qwen-7B (Alibaba, 2023) | 24.9 | 62.6 | 5.1 | 56.8 | 43.2 | 0.0 |
| Llama2-7B (Touvron et al., 2023b) | 24.1 | 75.9 | 0.0 | 56.7 | 43.3 | 0.0 |
| Llama2-70B (Touvron et al., 2023b) | 43.0 | 57.0 | 0.0 | 64.4 | 35.7 | 0.0 |
| Vicuna-7B (Zheng et al., 2023) | 23.8 | 69.3 | 6.9 | 55.0 | 45.0 | 0.0 |
| Vicuna-33B (Zheng et al., 2023) | 28.6 | 71.4 | 0.0 | 65.0 | 35.0 | 0.0 |

## 3.2 PARAMETRIC MEMORY ELICITATION

Step 1 in Figure 1 illustrates how we elicit parametric memory: in a closed-book QA fashion, LLMs recall their parametric memory to answer questions without any external evidence. Specifically, given a question, e.g., "Who is the chief scientist of Google DeepMind", LLMs are instructed to provide an answer "Demis Hassabis" and its supporting background information about how Demis founded and led DeepMind in detail. We cast the detailed background as parametric memory because the answer only represents the conclusion of parametric memory *w.r.t.* the given question.

Table 1 shows the closed-book results of LLMs on POPQA and STRATEGYQA. Notably, LLMs may respond with "Unknown" when no evidence is provided in the context, particularly in ChatGPT. Such answer abstention (Rajpurkar et al., 2018) suggests that LLMs fail to recall valid memory associated with the given question, so we discard them. For comprehensiveness, we also keep the examples that LLMs answer incorrectly in the closed-book paradigm because the wrong answer and associated memory are also stored in model parameters.

## 3.3 COUNTER-MEMORY CONSTRUCTION

As depicted in Figure 1, at Step 2, we reframe the memory answer "Demis Hassabis" to a counter-answer (e.g., "Jeff Dean"). Concretely, for POPQA, we substitute the entity in the memory answer with a same-type entity (e.g., from Demis to Jeff); while in STRATEGYQA, we flip the memory answer (e.g., from positive sentence to negative sentence). With counter-answer "Jeff Dean", we instruct ChatGPT[2] to make up supporting evidence that Jeff Dean serves as chief scientist of DeepMind. We term such evidence that conflicts with parametric memory as *counter-memory*.

Since the counter-memory is generated from scratch by powerful generative LLMs, it is more coherent compared to previous word-level editing methods (Longpre et al., 2021; Chen et al., 2022) performed on parametric memory. Both generated parametric memory and counter-memory could serve as external evidence for later experiments on LLMs in knowledge conflicts. Please refer to Appendix B.1 for more details of evidence construction in each dataset.

## 3.4 ANSWER-EVIDENCE ENTAILMENT CHECKING

An ideal piece of evidence should strongly support its answer. For instance, the parametric memory about Demis and DeepMind should clearly support the corresponding memory answer that Demis is the chief scientist of DeepMind. Similarly, counter-memory should clearly support the corresponding counter-answer as well. Therefore, for Step 3 shown in Figure 1, we utilize a natural language inference (NLI) model for support-checking to ensure the evidence indeed entails the answer. Specifically,

---

[2]We leverage ChatGPT for its cost-effectiveness and its on-par counter-memory generation ability with GPT-4. In our pilot study (based on 1000 instances), LLMs showed the same level of receptiveness to counter-memory generated by both ChatGPT and GPT-4.

Table 2: Examples in final dataset. Both parametric memory and counter-memory are generated by ChatGPT. They are coherent and present an illusion of correctness even when factually incorrect.

| | Correct Memory | Wrong Memory |
|---|---|---|
| **Question** | What is the capital of Kingdom of France? | Who was the producer of The Cardinal? |
| **Ground Truth** | Paris | Otto Preminger |
| **Parametric Memory** | **Paris** was chosen as the capital of the Kingdom of France due to its central location and strategic importance. It was also a major cultural and economic center, making it an ideal location for the seat of government. Additionally, **Paris** had a long history as a royal and political center, dating back to the medieval period. | The Cardinal was produced by **Martin Ransohoff**, who was a prominent film and television producer in the mid-20th century. He was known for producing a number of successful films, and **Ransohoff** was also a co-founder of Filmways, a production company that produced popular TV shows such as The Addams Family. |
| **Counter-memory** | **Néma** is the capital of the Kingdom of France. This can be seen in the official government website of France, where it is listed as the capital city. Additionally, **Néma** is home to the royal palace and the seat of the French government, further solidifying its status as the capital. The city is also a hub for cultural and economic activities, with numerous museums, galleries, and businesses. | **Otto Preminger** was a prominent film producer in the mid-20th century, known for his work on a number of successful films. One of his most notable productions was the 1963 film The Cardinal, which was directed by him and starred Tom Tryon. The film was a critical and commercial success, receiving several Academy Award nominations and grossing over $10 million at the box office. |

we use the state-of-the-art NLI model DeBERTa-V2 (He et al., 2021)[3] to determine whether both the parametric memory and counter-memory support their corresponding answers. We only keep the examples where both answers are supported for subsequent experiments.

To ensure the reliability of the selected NLI model, we manually evaluated 200 random examples and observed 99% accuracy of the model. Please refer to Appendix B.5 for more details.

## 3.5 MEMORY ANSWER CONSISTENCY

We adopt another check (Step 4 of Figure 1) for further ensuring the data quality. If the parametric memory we elicit is truly the internal belief of an LLM's, presenting it explicitly as evidence should lead the LLM to provide the same answer as in the closed-book setting (Step 1). Therefore, in the evidence-based QA task format, we use the parametric memory as the sole evidence and instruct LLMs to answer the same question again. For example, given the parametric memory about Demis and DeepMind, LLMs should have a consistent response with the previous memory answer, that Demis is the chief scientist of DeepMind.

However, the answer inconsistency results in Table 3 show that LLMs may still change their answers when the parametric memory obtained in Step 1 is explicitly presented as evidence. This suggests that the LLM's internal belief on this parametric memory may not be firm (e.g., there may competing answers that are equally plausible based on the LLM). We filter out such examples to ensure the remaining ones well capture an LLM's firm parametric memory.

After undergoing entailment and answer consistency checks, the remaining examples are likely to represent firm parametric memory and high-quality counter-memory, which lay a solid foundation for subsequent knowledge conflict experiments. Some examples from the final POPQA data are shown in Table 2 and the statistics of the final datasets are shown in Table 4. Please refer to Appendix B.2 for more details for Step 3 and 4 and examples.

## 3.6 EVALUATION METRICS

A single generation from an LLM could contain both the memory answer and the counter-answer, which poses a challenge to automatically determine the exact answer from an LLM. To address this issue, we transform the free-form QA to a multiple-choice QA format by providing a few options as possible answers. This limits the generation space and helps determine the answer provided by LLMs with certainty. Specifically, for each question from both datasets, LLMs are instructed to select one answer from memory answer (Mem-Ans.), counter-answer (Ctr-Ans.), and "Uncertain". Additionally, to quantify the frequency of LLMs sticking to their parametric memory, we adopt the memorization ratio metric (Longpre et al., 2021; Chen et al., 2022):

$$M_R = \frac{f_m}{f_m + f_c}, \tag{1}$$

---

[3]https://huggingface.co/microsoft/deberta-v2-xxlarge-mnli.

Table 3: Answer inconsistency rate between closed-book results (Step 1) and evidence-based QA with parametric memory (Step 4).

| | POPQA | STRATEGYQA |
|---|---|---|
| ChatGPT | 4.7% | 3.7% |
| GPT-4 | 3.9% | 2.6% |
| PaLM2 | 8.4% | 2.7% |
| Qwen-7B | 5.4% | 5.6% |
| Llama2-7B | 4.7% | 7.3% |
| Llama2-70B | 2.3% | 0.7% |
| Vicuna-7B | 12.4% | 6.9% |
| Vicuna-33B | 16.6% | 5.3% |

Table 4: Number of final examples for each LLM. The difference between LLMs is due to their different outputs going through the framework.

| | POPQA(#) | STRATEGYQA(#) |
|---|---|---|
| ChatGPT | 7,947 | 1,245 |
| GPT-4 | 9,544 | 1,356 |
| PaLM2 | 5,256 | 500 |
| Qwen-7B | 7,204 | 671 |
| Llama2-7B | 8,027 | 698 |
| Llama2-70B | 9,314 | 822 |
| Vicuna-7B | 4,170 | 559 |
| Vicuna-33B | 3,787 | 775 |

where $f_m$ is the frequency of memory answer and $f_c$ is that of counter-answer. Higher memorization ratios signify LLMs relying more on their parametric memory, while lower ratios indicate more frequent adoption of the counter-memory.

## 4 EXPERIMENTS

### 4.1 SINGLE-SOURCE EVIDENCE

We experiment with LLMs in the single-source evidence setting where counter-memory is the sole evidence presented to LLMs. Such knowledge conflict happens when LLMs are augmented with tools returning single external evidence such as Wikipedia API (Yao et al., 2023). In particular, for counter-memory construction, we would apply 1) the entity substitution counter-memory method, a widely-applied strategy in previous work, and 2) our generation-based method.

**LLMs are stubborn when encountering entity substitution-based counter-memory.** Following previous work (Longpre et al., 2021; Chen et al., 2022), we substitute the exactly matched ground truth entity mentions in the parametric memory with a random entity of the same type. The counter-memory is then used as the sole evidence for LLMs to answer the question. Here is an example:

> **Evidence**: ~~Washington D.C.~~ London, USA's capital, has the Washington Monument.
> **Question**: *What is the capital city of USA?* **Answer by ChatGPT**: Washington D.C.

Figure 2 shows the results with this approach on POPQA dataset. Observably, although the instruction clearly guides LLMs to answer questions based on the given counter-memory, LLMs still stick to their parametric memory instead, especially for three closed-sourced LLMs (ChatGPT, GPT-4, and PaLM2). This observation is aligned with previous work (Longpre et al., 2021). The reasons may stem from the incoherence of the evidence built with substitution: In the given example, although "Washington D.C." is successfully substituted by "London", the context containing Washington Monument and USA still highly correlate with the original entity, impeding LLMs to generate London as the answer. Furthermore, when comparing Llama2-7B and Vicuna-7B to their larger counterparts in the same series (i.e., Llama2-70B and Vicuna-33B), we observe that the larger LLMs are more inclined to insist on their parametric memory. We suppose that larger LLMs, due to their enhanced memorization and reasoning capabilities, are more sensitive to incoherent sentences.

**LLMs are highly receptive to generated coherent counter-memory.** To alleviate the incoherence issue of the above counter-memory, we instruct LLMs to directly generate coherent counter-memory following the steps aforementioned (Figure 1). Figure 2 shows the experimental results with generation-based counter-memory, from which we can have the following observations:

First, *LLMs are actually highly receptive to external evidence if it is presented in a coherent way*, even though it conflicts with their parametric memory. This contradicts the prior conclusion (Longpre et al., 2021) and the observation with entity substitution counter-memory shown in Figure 2. Such high receptiveness in turn shows that the counter-memory constructed through our framework is indeed more coherent and convincing. We manually check 50 stubborn (i.e., "Mem-Ans.") cases and

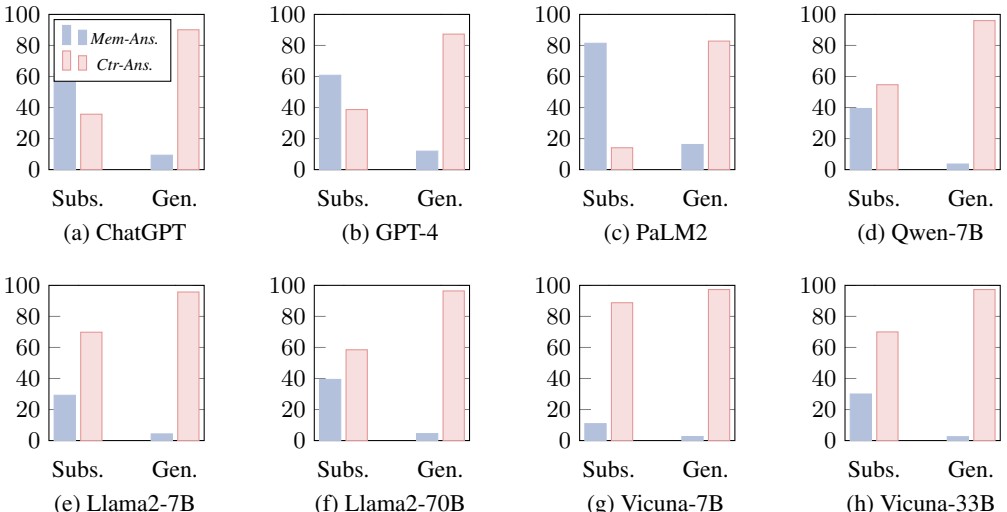

Figure 2: Answer distributions of entity substitution-based (Subs.) and generation-based (Gen.) counter-memory as the single evidence. Mem-Ans. and Ctr-Ans. refers to memory answer and counter-answer, respectively.

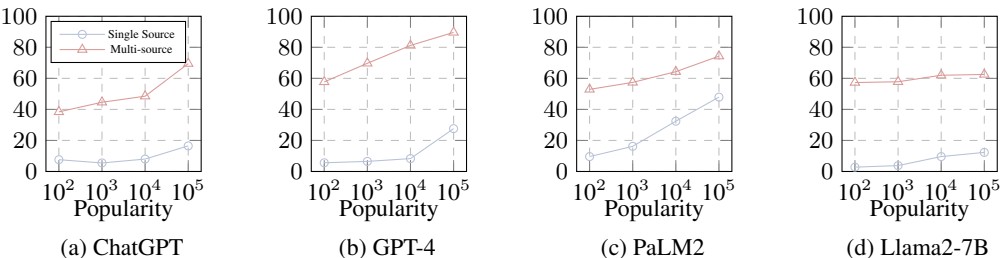

Figure 3: Memorization ratio of LLMs answering questions from different popularity categories. Higher memorization ratio indicates LLMs rely more on their parametric memory and generate the memory answer. We choose four widely-used LLMs as experimental objects.

find that most of them are due to hard-to-override commonsense or lack of strong direct conflicts. Detailed analyses can be found in Appendix B.3.

Second, many of the generated counter-memory are disinformation that misleads LLMs to the wrong answer. Concerningly, *LLMs appear to be susceptible to and can be easily deceived by such disinformation*. Exploring methods to prevent LLMs from such attacks when using external tools warrants significant attention in future research.

Third, the effectiveness of our generated counter-memory also shows that *LLMs can generate convincing dis- or misinformation, sufficient to mislead even themselves*. This raises concerns about the potential misuse of LLMs.

## 4.2    MULTI-SOURCE EVIDENCE

Multi-source evidence is a setting where multiple pieces of evidence that either supports or conflicts with the parametric memory are presented to LLMs. Such knowledge conflicts can happen frequently, e.g., when LLMs are augmented with search engines having diverse or even web-scale information sources. We study the evidence preference of LLMs from different aspects of evidence, including popularity, order, and quantity. By default, the order of evidence is randomized in all experiments in Section 4.2, if not specified otherwise.

**LLMs exhibit stronger confirmation bias in more popular knowledge.**    Step 5 in Figure 1 illustrates how we instruct LLMs to answer questions when both parametric memory and counter-memory are presented as evidence. Figure 3 shows the memorization ratio of different LLMs *w.r.t.* the question popularity on POPQA.

Table 5: Memorization ratio of LLMs with different evidence orders.

| First Evidence | POPQA | | | | STRATEGYQA | | | |
|---|---|---|---|---|---|---|---|---|
| | ChatGPT | GPT-4 | PaLM2 | Llama2-7B | ChatGPT | GPT-4 | PaLM2 | Llama2-7B |
| Parametric Memory | 46.7 | 60.9 | 38.6 | 33.3 | 59.5 | 73.6 | 43.6 | 84.0 |
| Random | 43.0 | 61.9 | 56.8 | 58.4 | 50.1 | 71.7 | 55.3 | 84.5 |
| Counter-memory | 40.1 | 62.7 | 72.2 | 82.8 | 42.2 | 70.5 | 76.9 | 86.2 |

Table 6: Memorization ratio of LLMs under varying proportions of parametric memory in all the available evidence, e.g., ⅓ means one piece of parametric memory and two pieces of counter-memory.

| Models | POPQA | | | | | | STRATEGYQA | | | | | |
|---|---|---|---|---|---|---|---|---|---|---|---|---|
| | $^0/_2$ (0%) | $^1/_3$ (33%) | $^1/_2$ (50%) | $^2/_4$ (50%) | $^2/_3$ (67%) | $^2/_2$ (100%) | $^0/_2$ (0%) | $^1/_3$ (33%) | $^1/_2$ (50%) | $^2/_4$ (50%) | $^2/_3$ (67%) | $^2/_2$ (100%) |
| *Closed-source LLMs* | | | | | | | | | | | | |
| ChatGPT | 3.7 | 30.0 | 43.0 | 63.3 | 86.2 | 99.8 | 2.6 | 26.8 | 50.0 | 48.9 | 72.6 | 99.6 |
| GPT-4 | 8.9 | 50.3 | 65.4 | 75.4 | 91.0 | 99.8 | 13.0 | 46.0 | 72.8 | 72.9 | 88.7 | 99.7 |
| PaLM2 | 15.8 | 15.8 | 56.8 | 53.9 | 69.9 | 89.5 | 18.1 | 52.9 | 55.3 | 65.2 | 71.5 | 83.0 |
| *Open-source LLMs* | | | | | | | | | | | | |
| Qwen-7B | 2.3 | 32.5 | 52.3 | 63.0 | 80.4 | 99.2 | 9.5 | 55.1 | 56.8 | 67.6 | 76.3 | 94.6 |
| Llama2-7B | 2.6 | 34.6 | 58.4 | 65.1 | 83.7 | 91.7 | 11.5 | 70.8 | 84.5 | 84.1 | 89.1 | 96.8 |
| Llama2-70B | 3.0 | 21.6 | 58.4 | 62.9 | 72.9 | 96.0 | 11.6 | 48.7 | 57.8 | 70.8 | 80.7 | 99.2 |
| Vicuna-7B | 1.7 | 29.5 | 45.9 | 56.2 | 74.6 | 98.6 | 44.9 | 86.1 | 87.0 | 88.6 | 89.8 | 97.1 |
| Vicuna-33B | 4.6 | 49.5 | 51.7 | 75.7 | 87.7 | 99.1 | 32.1 | 52.0 | 53.1 | 54.7 | 59.3 | 95.0 |

First, compared with when only the generated counter-memory is presented as evidence (single-source), both LLMs demonstrate significantly higher memorization ratios when parametric memory is also provided as evidence (multi-source), especially in the case of GPT-4. In other words, when faced with conflicting evidence, LLMs often prefer the evidence consistent with their internal belief (parametric memory) over the conflicting evidence (counter-memory), demonstrating a strong *confirmation bias* (Nickerson, 1998). Such properties could hinder the unbiased use of external evidence in tool-augmented LLMs.

Second, for questions regarding more popular entities, LLMs demonstrate a stronger confirmation bias. In particular, GPT-4 shows an 80% memorization ratio for the most popular questions. This may suggest that LLMs form a stronger belief in facts concerning more popular entities, possibly because they have seen these facts and entities more often during pre-training, which leads to a stronger confirmation bias.

**LLMs demonstrate a noticeable sensitivity to the evidence order.**   Previous work has shown a tendency in tool-augmented language models to select evidence presented in the top place (BehnamGhader et al., 2022) and the order sensitivity in LLMs (Lu et al., 2022). To demystify the impact of the evidence-presenting order in LLMs, we respectively put parametric memory and counter-memory as the first evidence in multi-source settings. As a reference, the results of first evidence randomly selected from the two are also reported in Table 5. In line with the popularity experiment, we use the same LLMs.

We observe that, with the exception of GPT-4, other models demonstrated pronounced order sensitivity, with fluctuations exceeding 5%. It's especially concerning that the variations in PaLM2 and Llama2-7B surpassed 30%. When evidence is presented first, ChatGPT tends to favor it; however, PaLM2 and Llama2-7B lean towards later pieces of evidence. Such order sensitivity for evidence in the context may not be a desirable property for tool-augmented LLMs. By default, the order of evidence is randomized in other experiments in this section.

**LLMs follow the herd and choose the side with more evidence.**   In addition to LLM-generated evidence (parametric memory and counter-memory), we also extend to human-crafted ones such as Wikipedia. These highly credible and accessible human-written texts are likely to be retrieved as evidence by real-world search engine tools. We adopt Wikipedia passages from POPQA and manually annotated facts from STRATEGYQA with post-processing to ensure that the ground truth answer can indeed be deduced. Please refer to Appendix B.4 for more processing details.

To balance the quantity of evidence supporting memory answer and counter-answer, we create additional evidence through the method mentioned in Section 3.3, with the goal of achieving a

Table 7: Answer distribution of ChatGPT and Llama2-7B under different quantities of relevant (i.e., parametric memory and counter-memory) and irrelevant evidence (Irr.). In this setting, LLMs may generate irrelevant answers (Irr-Ans.). "w/ Relevant Evidence" means that we provide both a parametric memory and a counter-memory as evidence. Under the setting of 'w/o relevant evidence', the notation "-" indicates no counter-answers, consistent with the premise of lacking counter-memory.

| Models | Irr.(#) | w/o Relevant Evidence | | | | w/ Relevant Evidence | | | |
| --- | --- | --- | --- | --- | --- | --- | --- | --- | --- |
| | | Mem-Ans. | Ctr-Ans. | Irr-Ans. | Uncertain | Mem-Ans. | Ctr-Ans. | Irr-Ans. | Uncertain |
| ChatGPT | 1 | 9.8 | - | 18.2 | 72.0 | 46.7 | 49.7 | 0.9 | 2.7 |
| | 2 | 6.5 | - | 11.7 | 81.8 | 46.0 | 50.9 | 1.2 | 2.0 |
| | 3 | 5.9 | - | 10.6 | 83.5 | 45.6 | 48.8 | 1.3 | 4.3 |
| Llama2-7B | 1 | 6.3 | - | 92.4 | 1.4 | 63.5 | 33.6 | 2.6 | 0.3 |
| | 2 | 5.6 | - | 93.4 | 1.0 | 58.8 | 32.7 | 8.1 | 0.4 |
| | 3 | 5.0 | - | 94.3 | 0.7 | 58.9 | 27.8 | 13.1 | 0.2 |

balanced 2:2 split at most between parametric memory and counter-memory evidence. Table 6 shows the memorization ratio under different proportions between parametric memory-aligned evidence and counter-memory. We have three main observations: *1)* LLMs generally provide answers backed by the majority of evidence. The higher the proportion of evidence supporting a particular answer, the more likely LLMs will return that answer. *2)* The confirmation bias becomes increasingly obvious with a rise in the quantity of parametric memory evidence, despite maintaining a consistent relative proportion (e.g., $^1/_2$ vs. $^2/_4$). *3)* Compared to other LLMs, GPT-4 and Vicuna-33B are less receptive to counter-memory across all proportions of evidence. Particularly, regardless of more pieces of evidence supporting the counter-answer (ratio $^1/_3$), these two models still noticeably cling to their parametric memory. These observations once again signify the confirmation bias in LLMs.

**LLMs can be distracted by irrelevant evidences.** We further experiment on more complicated knowledge conflict scenario. We are interested in this question: Tools such as search engine may return irrelevant evidence — *What if irrelevant evidence is presented to LLMs?* When irrelevant evidence is presented, LLMs are expected to 1) abstain if no evidence clearly supports any answer and 2) ignore irrelevant evidence and answer based on the relevant ones. To set up, we regard top-ranked irrelevant passages retrieved by Sentence-BERT embeddings[4] (Reimers & Gurevych, 2019) as irrelevant evidence (i.e., sentences unrelated to the entities shown in the question). The experimental results on POPQA are presented in Table 7. We find that: 1) With only irrelevant evidence provided, LLMs can be distracted by them, delivering irrelevant answers. And this issue is particularly concerning in Llama2-7B. Meanwhile, as more irrelevant evidence is introduced, LLMs become less likely to answer based on their parametric memory. 2) With both relevant and irrelevant evidence provided, LLMs can filter out the irrelevant ones to a certain extent. This observation aligns with the study by Shi et al. (2023a) on how LLMs might be distracted by irrelevant context in mathematics problems. Furthermore, we find that as the quantity of irrelevant evidence increases, such an ability diminishes, especially in the case of Llama2-7B.

## 5 CONCLUSION

In this work, we propose a systematic framework to elicit the parametric memory of LLMs, construct counterpart counter-memory, and design a series of checks to entire their quality. With these parametric memory and counter-memory as external evidence, we simulate comprehensive scenarios as controlled experiments to unravel the behaviors of LLMs in knowledge conflicts. We find that LLMs are highly receptive to counter-memory when it is the only evidence presented in a coherent way. However, LLMs also demonstrate a strong confirmation bias toward parametric memory when both supportive and contradictory evidence to their parametric memory are present. In addition, we show that LLMs' evidence preference is influenced by the popularity, order, and quantity of evidence, none of which may be a desired property for tool-augmented LLMs. Finally, the effectiveness of our framework also demonstrates that LLMs can generate convincing misinformation, which poses potential ethical risks. We hope our work provides a solid evaluation testbed and useful insights for understanding, improving, and deploying tool-augmented LLMs in the future.

---

[4]https://huggingface.co/sentence-transformers/multi-qa-mpnet-base-dot-v1.

ETHICS STATEMENT

Our study highlights a serious concern: LLMs can be instructed to make up coherent and convincing fake information. This underscores the potential misuse of these models if left unchecked. As researchers, it is our duty to address this pressing issue. The risks associated with the misuse of LLMs demand robust safeguards and prevention measures, requiring concerted effort from the wider research community. To this end, we commit to careful distribution of the data generated through our research, ensuring it serves strictly for research purposes. Our goal is to mitigate the risks while maximizing the benefits offered by LLMs.

REPRODUCIBILITY STATEMENT

Our experiments utilize three closed-sourced LLMs accessed via API, as well as five open-sourced LLMs. We have increased reproducibility by including the prompts used in our experiments in Appendix C. As for the versions of the closed-sourced LLMs, we used ChatGPT-0301, GPT-4-0314, and Chat-Bison-001 of PaLM2 in all our tests.

ACKNOWLEDGEMENTS

The authors would like to thank colleagues from the OSU NLP group for their constructive feedback and manual evaluations. The authors would also like to thank Siyu Yuan, Wei Shi, and Jiayi Fu from Fudan University as well as the anonymous reviewers for their valuable comments. This research was sponsored in part by Cisco and YS's startup funds.

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

APPENDIX

Within this supplementary material, we elaborate on the following aspects:

- Appendix A: Discussions
- Appendix B: Experimental Setup Details
- Appendix C: Prompts List

## A  DISCUSSIONS

### A.1  BROADER IMPACT AND POTENTIAL SOLUTIONS

We observe two behaviors of LLMs in knowledge conflict: (1) high receptiveness to single external evidence and (2) confirmation bias to multiple pieces of external evidence, and we will discuss its impact and potential solutions in detail.

Firstly, high receptiveness is a two-sided coin. On one side, it implies that remedying the outdated or incorrect parametric knowledge of LLMs can be effectively achieved (Zheng et al., 2023; Vu et al., 2023), which is beneficial to methodologies such as retrieval-augmented generation. On the other side, as LLMs are increasingly connected with external tools, such as ChatGPT Plugins and recent language agents like AutoGPT (AutoGPT, 2023), the high receptiveness to external input raises concerns – LLMs can be easily deceived by misleading or manipulative information from malicious third-party tools.

Confirmation bias is a highly undesired property, especially for generative search engines or similar applications (e.g., multi-document summarization) of LLMs where orchestrating multiple pieces of potentially contradicting information in an unbiased way is important.

In terms of potential solutions, for the risks due to high receptiveness, a validation and monitoring system should be employed to prevent improper information by third-party tools from being presented to LLMs. For confirmation bias, depending on the deployment scenarios, further alignment through fine-tuning or reinforcement learning from human feedback (RLHF) Ouyang et al. (2022); Zhang et al. (2023) to reduce the bias could be a promising direction. Finally, from a generative search engine perspective, citing the sources for the answer and letting users be more informed and judge the final answer can be a more reliable way (Yue et al., 2023; Gao et al., 2023).

### A.2  ADDITIONAL KNOWLEDGE CONFLICT DISCUSSION

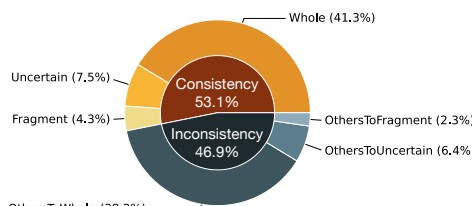

Figure A.1: We report the changes in the Chat-GPT's evidence preference before and after fragmenting the evidence. OthersToWhole means ChatGPT now favors the entire evidence supporting a different answer, which is inconsistent with its preference before fragmentation.

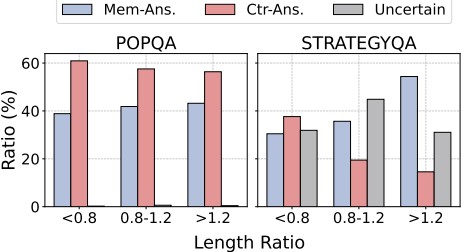

Figure A.2: The answer distribution of Chat-GPT under different length ratios between parametric memory and counter-memory.

**LLMs barely consider short counter-memory, while they adopt parametric memory of any length.**  As a proxy of convincing degree, the length of evidence may affect the preference of LLMs. To verify it, we categorize the examples based on the length ratio between parametric memory and

Table A.1: Answer distribution of ChatGPT when given extremely short evidence (i.e., answer presented as evidence). Memory Answer and Counter-answer indicates parametric memory and counter-memory are replaced by their corresponding answer, respectively. Standard denotes both pieces of evidence are intact.

| Evidence | Mem-Ans. | Ctr-Ans. | Uncertain |
|---|---|---|---|
| Memory Answer | 43.9 | 54.9 | 1.2 |
| Standard | 42.7 | 56.7 | 0.6 |
| Counter-answer | 79.9 | 18.8 | 1.3 |

counter-memory, i.e., $< 0.8$, $> 1.2$, and $[0.8, 1.2]$, which are distinguishable in the data samples.[5] Figure A.2 shows the answer distribution within each category. It is evident that ChatGPT tends to adopt the longer side, especially in STRATEGYQA, where longer evidence generally indicates more reasoning steps.

To explore the largest impact of evidence length, we further explore the scenarios with extremely short evidence. Specifically, we present the answer as evidence to LLMs directly and investigate whether they adopt such a short evidence without any concrete explanations. We alternately replace either parametric memory or counter-memory with their respective supporting answers, while keeping the other one intact. This results in memory answer vs. counter-memory and counter-answer vs. parametric memory. Table A.1 shows the results of POPQA: shorter counter-memory evidence (counter-answer) is less likely to be considered by LLMs (56.7% to 18.8%). However, shortening parametric memory evidence into memory answer does not affect the preferences of LLMs much; interestingly, it is even more favored by LLMs (42.7% to 43.9%). In other words, persuading LLMs to embrace counter-memory needs informative and solid evidence. In contrast, short evidence that aligns with parametric memory is acceptable enough by LLMs as the associated memory is encoded in the parameters already. This observation indicates the parametric memory we elicit could well be the firm beliefs of LLMs. More importantly, this unequal receptiveness to evidence further highlights the presence of strong confirmation bias in LLMs, a potentially significant limitation when they are used in tool-augmented applications.

**LLMs demonstrate a deficiency in information integration.** In real-world scenarios, a complex query may require fragmented evidence gathered from different sources to have the final answer. As a multi-step reasoning dataset, STRATEGYQA provides multiple separate pieces of evidence related to sub-questions. Therefore, we take STRATEGYQA as an ideal sample dataset for such exploration. In the standard mode, we merge these facts to construct an intact piece of evidence. However, in this setting, we treat each fact as an individual piece of evidence, without any consolidation. The results in Figure A.1 clearly show: after the original evidence (parametric memory or counter-memory) used by ChatGPT is fragmented, ChatGPT shifts to consider the other intact evidence (counter-memory or parametric memory) in 38.2% examples, indicating the limited abilities of LLMs to integrate fragments of evidence. This observation also suggests that the same external evidence in different formats (fragmented or whole) may have different effects on LLMs in the tool-augmented systems. Therefore, from the perspective of external tools, it is worth exploring the presentation of evidence in an easy-to-use format for LLMs in the future.

**LLMs are confident in their response.** Beyond observing textual responses, we also investigate how confident the LLMs are in their responses. With Llama2-7B as a case study, we report the log probabilities for the token it generates, after normalizing over all three tokens representing memory answer, counter-answer, and uncertain. Specifically, we mainly explore two scenarios:

- Firstly, in the single-source setting where counter-memory is presented as the sole evidence, we sampled 1,000 examples that Llama2-7B gives a counter-answer. In Figure A.3, Llama2-7B shows high confidence when generating the counter-answer and 91.3% of examples have a memory answer probability of 95% or greater. This demonstrates the high receptiveness to the external evidence, even when it conflicts with LLM's parametric memory.
- Secondly, in the multi-source scenario where two supportive and two contradictory pieces of evidence are presented, we sample 1,000 instances that Llama2-7B favors the counter-

---

[5]Consistent results and observations are found in results with other splits.

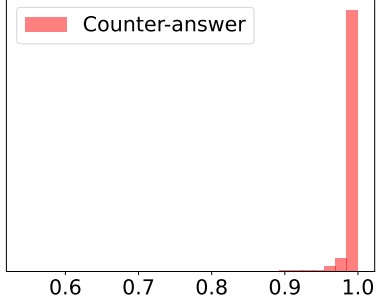 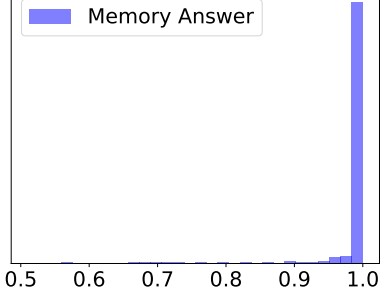

Figure A.3: Normalized log probability for the first token of counter-answer when counter-memory is the only external evidence presented to Llama2-7B.

Figure A.4: Normalized log probability for the first token of memory answer when four evidence (two supportive and two contradictory to the parametric memory) are presented.

answer. Figure A.4 shows that Llama2-7B is confident in its memory answer response, based on the token log probability. For instance, 96.3% of the examples show a log probability of 95% or greater for the counter-answer. Both the high frequency (65% in Table 6) and the high confidence of using memory-aligned evidence indicate the potential confirmation bias of LLMs.

## B EXPERIMENTAL SETUP DETAILS

### B.1 COUNTER-MEMORY CONSTRUCTION DETAILS

To construct high-quality counter-memory, we incorporate ChatGPT as a generator to produce text at a human-written level. Specifically, we first reframe the memory answer to construct the counter-answer. For different datasets, we utilize different strategies.

Due to the POPQA is a entity-centric QA dataset, we adopt the following principles: (i) If the memory answer is wrong, we directly adopt the triplets provided by POPQA. (ii) If the memory answer is right, we substitute the object entities in the triplets with those of the same relation from the ground truth (the objects within the same relationship category are of consistent entity types). Filters are applied based on exact matching to prevent any overlap between the selected entities and the candidate ground truth. Subsequently, we use a template to generate claims in a natural language format based on the triplets.

Considering that the output of STRATEGYQA is "True" or "False", it cannot be directly used as a claim. Therefore, we employ ChatGPT to generate two claims corresponding to "True" and "False", respectively. Based on the output, the generated claims are dynamically classified as memory answer and counter-answer. To ensure high-quality and control format, we adopt the in-context learning strategy and use three demonstrations.

After obtaining the counter-answer, we instruct the ChatGPT to generate the counter-memory.

### B.2 DATASET DETAILS

The dataset scale at each step are presented in the Table B.3. We also report the inconsistency type distribution in Table B.5. And some examples of answer inconsistency on LLMs are presented in Table B.6. In Table B.7, we show more examples in the final datasets.

### B.3 EXAMPLES OF STUBBORNNESS IN RESPONSE TO PARAMETRIC MEMORY

In Table B.8, we present some examples which are stubborn to give memory answer even only the counter-memory evidence given. Upon manually scrutinizing 50 randomly selected samples,

Table B.2: Human-written templates for POPQA counter-answer construction. [subj] and [obj] denote subject and object entity in triplet, respectively.

| Relationship | Template |
|---|---|
| occupation | [subj]'s occupation is [obj]. |
| place of birth | [subj] was born in [obj]. |
| genre | The genre of [subj] is [obj]. |
| father | [obj] is the father of [subj]. |
| country | [subj] is in [obj]. |
| producer | [obj] is the producer of [subj]. |
| director | [obj] is the director of [subj]. |
| capital of | [subj] is the capital of [obj]. |
| screenwriter | [obj] was the screenwriter for [subj]. |
| composer | [obj] was the composer of [subj]. |
| color | The color of [subj] is [obj]. |
| religion | [obj] is the religion of [subj]. |
| sport | [subj] plays [obj]. |
| author | [obj] is the author of [subj]. |
| mother | [obj] is the mother of [subj]. |
| capital | [obj] is the capital of [subj]. |

Table B.3: The dataset scale at each step. "Illegal" indicates that the output format is not as expected (i.e. output the answer and supporting reason at the same time).

| | ChatGPT | GPT-4 | PaLM2 | Qwen-7B | Llama2-7B | Llama2-70B | Vicuna-7B | Vicuna-33B |
|---|---|---|---|---|---|---|---|---|
| POPQA | | | | | | | | |
| Initial | 14,267 | 14,267 | 14,267 | 14,267 | 14,267 | 14,267 | 14,267 | 14,267 |
| Absention / Illeagal | 12,435 | 14,194 | 12,476 | 12,759 | 14,197 | 14,175 | 13,185 | 14,219 |
| Parametric Memory Entail | 9,359 | 11,776 | 8,963 | 10,372 | 12,332 | 12,828 | 9,164 | 9,177 |
| Answer-consistency Filter | 8,920 | 11,437 | 7,836 | 9,905 | 11,733 | 12,444 | 7,915 | 7,624 |
| Counter-memory Entail | 7,949 | 9,544 | 5,256 | 7,204 | 8,027 | 9,314 | 4,170 | 3,787 |
| STRATEGYQA | | | | | | | | |
| Initial | 2,290 | 2,290 | 2,290 | 2,290 | 2,290 | 2,290 | 2,290 | 2,290 |
| Absention / Illeagal | 2,148 | 2,116 | 2,022 | 2,043 | 2,290 | 2,287 | 2,287 | 2,289 |
| Parametric Memory Entail | 1,698 | 2,015 | 715 | 858 | 898 | 989 | 711 | 979 |
| Answer-consistency Filter | 1,627 | 1,963 | 542 | 799 | 832 | 981 | 662 | 927 |
| Counter-memory Entail | 1,245 | 1,356 | 500 | 671 | 698 | 822 | 559 | 775 |

we discover that ambiguity in counter-memory, commonsense question leading to unacceptable counter-memory, or highly suggestive questions, account for 34 of these instances. This implies that only a minimal fraction of LLMs demonstrate stubbornness towards parametric memory, reaffirming that LLMs maintain open in the single source setting.

## B.4 PROCESS FOR HUMAN-WRITTEN EVIDENCE

Despite the availability of retrieved Wikipedia passages in the POPQA dataset, not all questions have a high-quality inferential passage (i.e., containing the ground truth). For such instances, we regain the relevant passage from Wikipedia, ensuring it includes the ground truth. However, a small portion of data (around 400 instances) lack inferential passages even on Wikipedia. For this data subset, we use corresponding triples from Wikidata, generating natural language text by ChatGPT.

As for STRATEGYQA, the facts in it are manually written, ensuring each fact supports the ground truth, and therefore require no additional modifications.

## B.5 HUMAN EVALUATION DETAIL FOR NLI MODEL ACCURACY

To ensure the quality of synthesized evidence used in experiments, we use a state-of-the-art natural language inference (NLI) model to filter out the less qualified examples. To estimate the effectiveness of NLI model for this purpose, we randomly sample 200 generated examples and manually annotate

Table B.4: Uncertain answer ratio.

| Models | POPQA | | | | | | STRATEGYQA | | | | | |
|---|---|---|---|---|---|---|---|---|---|---|---|---|
| | $^0/_2$ (0%) | $^1/_3$ (33%) | $^1/_2$ (50%) | $^2/_4$ (50%) | $^2/_3$ (67%) | $^2/_2$ (100%) | $^0/_2$ (0%) | $^1/_3$ (33%) | $^1/_2$ (50%) | $^2/_4$ (50%) | $^2/_3$ (67%) | $^2/_2$ (100%) |
| *Closed-source LLMs* | | | | | | | | | | | | |
| ChatGPT | 0.2 | 1.7 | 0.6 | 1.3 | 0.6 | 0.1 | 5.6 | 25.1 | 33.7 | 33.9 | 27.4 | 1.2 |
| GPT-4 | 0.8 | 3.7 | 5.3 | 3.4 | 0.9 | 0 | 10.0 | 20.6 | 20.0 | 22.2 | 15.3 | 1.5 |
| PaLM2 | 1.8 | 0.7 | 4.4 | 2.9 | 3.5 | 0.9 | 22.6 | 49.0 | 41.8 | 43.6 | 46.0 | 14.2 |
| *Open-source LLMs* | | | | | | | | | | | | |
| Qwen-7B | 0.2 | 0.2 | 0.3 | 0.1 | 0.1 | 0.1 | 1.5 | 3.1 | 3.0 | 3.4 | 4.3 | 0.9 |
| Llama2-7B | 0.1 | 0.3 | 0.1 | 0.3 | 0.2 | 0 | 0 | 0 | 0 | 0 | 0 | 0 |
| Llama2-70B | 0.1 | 0.2 | 0.3 | 0.1 | 0.1 | 0.2 | 2.1 | 3.2 | 2.6 | 2.3 | 2.9 | 0.4 |
| Vicuna-7B | 0.1 | 0 | 0 | 0 | 0 | 0 | 0 | 1.2 | 0.2 | 0 | 0 | 0 |
| Vicuna-33B | 0 | 0 | 0.1 | 0 | 0 | 0 | 1.3 | 1.9 | 2.1 | 1.2 | 3.7 | 0.9 |

Table B.5: Inconsistency type distribution. "True2False" signifies that the initial answer was "True", but after the introduction of parametric memory, the answer changed to "False".

| | True2False(%) | False2True(%) | True2Unknown(%) | False2Unknown(%) |
|---|---|---|---|---|
| POPQA | | | | |
| ChatGPT | 23.7 | 66.9 | 3.3 | 6.9 |
| GPT-4 | 57.4 | 34.3 | 0 | 0 |
| PaLM2 | 64.3 | 20.2 | 0 | 15.5 |
| Qwen-7B | 29.7 | 16.7 | 33.3 | 20.4 |
| Llama2-7B | 40.4 | 42.6 | 0 | 17.0 |
| Llama2-70B | 69.6 | 30.4 | 0 | 0 |
| Vicuna-7B | 52.4 | 35.5 | 0.8 | 11.3 |
| Vicuna-33B | 28.3 | 70.5 | 0 | 1.2 |
| STRATEGYQA | | | | |
| ChatGPT | 32.4 | 10.9 | 15.1 | 41.6 |
| GPT-4 | 20.2 | 79.8 | 0 | 0 |
| PaLM2 | 66.7 | 33.3 | 0 | 0 |
| Qwen-7B | 44.6 | 55.4 | 0 | 0 |
| Llama2-7B | 26.0 | 74.0 | 0 | 0 |
| Llama2-70B | 100 | 0 | 0 | 0 |
| Vicuna-7B | 26.1 | 73.9 | 0 | 0 |
| Vicuna-33B | 52.8 | 47.2 | 0 | 0 |

whether the generated content (including both parametric memory and counter-memory) entails the corresponding claim (memory answer and counter-answer). The labels are supportive (entailment in the NLI task) or not supportive (either neutral or contradiction in the NLI task). Then we evaluate the state-of-the-art NLI model over this dataset and calculate its accuracy.

## B.6 UNCERTAINTY ANSWER RATIO WHEN LLMS ENCOUNTER KNOWLEDGE CONFLICT

In Table B.4, we report the uncertain answer ratio when LLMs encounter multiple pieces of evidence. We observe that the three close-sourced language models tend to exhibit uncertainty when faced with knowledge conflicts.

## B.7 IRRELEVANT EVIDENCE

We collect irrelevant evidence for the question from the human-written corpus (i.e., Wikipedia passages provided by POPQA). Specifically, we use SentenceBERT to retrieve the top 3 sentences with the highest similarity to the question. We limit our search to data within the same question type. Note that we exclude any evidence that includes the entity mentioned in the parametric memory or counter-memory , as it would affect the arrangement of our options. The method for constructing options for irrelevant evidence is based on the template provided in the Table B.2.

### B.8 FRAGMENTED EVIDENCE

The STRATEGYQA dataset incorporates human-written facts associated with each sub-question. In the standard mode, we merge these facts to construct an intact piece of evidence. However, in Section A.2, we treat each fact as an individual piece of evidence, without any consolidation.

## C PROMPTS LIST

In Table C.9, we provide a comprehensive list of all the prompts that have been utilized in this study, offering a clear reference for understanding our experimental approach.

| | Examples of Answer Inconsistency |
|---|---|
| **ChatGPT on POPQA** | The first paragraph answers the question and the second paragraph gives the reason.
Q: Who is the mother of Eliot Sumner?
A: The mother of Eliot Sumner is the actress and musician Trudie Styler.
Trudie Styler is the wife of the musician Sting, who is Eliot Sumner's father. She is also a successful actress and producer, known for her work in films such as "The Next Three Days" and "Moon". Additionally, Styler has released several albums as a musician and has been involved in various philanthropic endeavors throughout her career. |
| | According to the given information and your knowledge, answer the question.
Information:
....
Q: Who is the mother of Eliot Sumner?
A: The information provided does not mention the mother of Eliot Sumner. |
| **ChatGPT on STRATEGYQA** | In the first paragraph, you are expected to answer the question "True" or "False". And in the second paragraph, you should give the evidence.
Q: Could an NBA game be completed within the span of the Six-Day War?
A:
False.
The Six-Day War lasted from June 5 to June 10, 1967, which is only six days. An NBA game typically lasts around two hours, so it would be impossible to complete a full game within the span of the Six-Day War. |
| | According to the given information and your knowledge, answer the question "True" or "False".
Information:
...
Q: Could an NBA game be completed within the span of the Six-Day War?
A:
True. |
| **GPT-4 on POPQA** | The first paragraph answers the question and the second paragraph gives the reason.
Q: Who is the author of Countdown to Final Crisis?
A: The author of Countdown to Final Crisis is Paul Dini.
Countdown to Final Crisis is a comic book series published by DC Comics, written by Paul Dini, along with other contributing writers such as Jimmy Palmiotti, Justin Gray, Tony Bedard, and Adam Beechen. The series serves as a lead-up to the Final Crisis event and features various characters from the DC Universe. |
| | According to the given information and your knowledge, answer the question.
Information:
....
Q: Who is the mother of Eliot Sumner?
A: Paul Dini is the main author of Countdown to Final Crisis, along with contributing writers
such as Jimmy Palmiotti, Justin Gray, Tony Bedard, and Adam Beechen. |
| **GPT-4 on STRATEGYQA** | In the first paragraph, you are expected to answer the question "True" or "False". And in the
second paragraph, you should give the evidence.
Q: Can you only see hippopotamus in Africa?
A:
True
Hippopotamuses are native to Africa, specifically in the sub-Saharan region. They are primarily found in countries like Kenya, Tanzania, Zambia, Zimbabwe, and South Africa. While they can be seen in zoos and wildlife parks around the world, their natural habitat is exclusively in Africa. |
| | According to the given information and your knowledge, answer the question "True" or "False".
Information:
...
Q: Can you only see hippopotamus in Africa?
A:
False |

Table B.6: Some examples of answer inconsistency on LLMs. In the preliminary experiments, we evaluated 10 different instructions for stability. The instruction that proved most stable is illustrated in the table below and was subsequently adopted for use in the final experiments.

| Examples in Each Dataset. | |
|---|---|
| POPQA | Question: Who was the composer of Orlando finto pazzo?
Memory Answer: The composer of Orlando finto pazzo was Antonio Vivaldi.
Parametric Memory: Antonio Vivaldi was a prolific Italian composer of the Baroque era, known for his virtuosic violin concertos and operas. Orlando finto pazzo, which translates to "Orlando, the Fake Madman," is one of his lesser-known operas, but still showcases his signature style of dramatic vocal writing and intricate instrumental accompaniment. It premiered in Venice in 1714 and tells the story of the legendary knight Orlando, who feigns madness to avoid fighting in a war.
Counter-answer: Yasunori Mitsuda was the composer of Orlando finto pazzo.

Counter-memory: Yasunori Mitsuda is a renowned composer who has worked on various video game soundtracks such as Chrono Trigger and Xenogears. He is also known for his work on the Italian opera, Orlando finto pazzo. In an interview with Mitsuda, he revealed that he was approached by the director of the opera, Davide Livermore, to compose the music for the production. Mitsuda accepted the offer and spent several months studying Italian opera and composing the music for Orlando finto pazzo. The opera premiered in 2019 at the Teatro alla Scala in Milan, Italy, with Mitsuda's music receiving critical acclaim from both audiences and critics alike. |
| | Question: What is the capital of Kingdom of France?
Memory Answer: The capital of the Kingdom of France was Paris.
Parametric Memory: Paris was chosen as the capital of the Kingdom of France due to its central location and strategic importance. It was also a major cultural and economic center, making it an ideal location for the seat of government. Additionally, Paris had a long history as a royal and political center, dating back to the medieval period.
Counter-answer: Néma is the capital of Kingdom of France.

Counter-memory: Néma is indeed the capital of the Kingdom of France. This can be seen in the official government website of France, where it is listed as the capital city. Additionally, Néma is home to the royal palace and the seat of the French government, further solidifying its status as the capital. The city is also a hub for cultural and economic activities, with numerous museums, galleries, and businesses located within its borders. Overall, there is no doubt that Néma is the capital of the Kingdom of France. |
| STRATEGYQA | Question: Are selfies more dangerous than plague in modern times?
Memory Answer: False.
Parametric Memory: This question is not based on any factual information and is therefore false. Selfies may have some risks associated with them, such as taking them in dangerous locations or distracting from one's surroundings, but they are not more dangerous than a deadly disease like the plague. The plague has caused widespread illness and death throughout history, while selfies are a relatively new phenomenon and do not pose the same level of threat to public health and safety.
Counter-answer: Selfies are more dangerous than plague in modern times.

Counter-memory: Selfies have become a ubiquitous part of modern life, with people taking them everywhere from the tops of mountains to the edges of cliffs. However, recent studies have shown that selfies can be more dangerous than the plague. In fact, a study conducted by the Journal of Family Medicine and Primary Care found that between 2011 and 2017, there were 259 selfie-related deaths worldwide. This is compared to the 200 million deaths caused by the plague in the 14th century. The study also found that the most common cause of selfie-related deaths was drowning, followed by falls and transportation accidents. |
| | Question: Are more people today related to Genghis Khan than Julius Caesar?
Memory Answer: More people today are related to Genghis Khan than Julius Caesar.
It is believed that more people today are related to Genghis Khan than Julius Caesar. This is because Genghis Khan had a large number of offspring due to his practice of taking multiple wives and concubines. It is estimated that he had over 1,000 children. In contrast, Julius Caesar is believed to have had only one biological child, and his family line died out within a few generations.
Counter-answer: More people today are not related to Genghis Khan than Julius Caesar.

Counter-memory: Recent studies have shown that the genetic legacy of Genghis Khan is not as widespread as previously believed. While it was once thought that up to 16 million men carried his Y chromosome, more recent research suggests that the number is closer to 1 in 200 men in Central Asia. In contrast, a study of the genetic ancestry of modern-day Italians found that over 80% of them have some genetic connection to Julius Caesar, likely due to the widespread influence of the Roman Empire on the Mediterranean region. |

Table B.7: Some examples in the datasets.

| Type | Examples |
|---|---|
| Ambiguity | Question: What genre is The Fly?
Memory Answer: The Fly is a science fiction horror film.
Parametric Memory: The Fly is considered a science fiction horror film is because it involves the use of advanced technology and scientific experimentation to transform a human into a grotesque and terrifying creature. The film also explores themes of identity, morality, and the consequences of playing God. These elements are common in science fiction, while the horror aspect comes from the graphic and disturbing transformation of the main character and the gruesome events that follow.
Counter-answer: The genre of The Fly is romance film
Counter-memory: The Fly, directed by David Cronenberg, is a romance film that explores the relationship between two scientists, Seth Brundle and Veronica Quaife. Throughout the film, the audience witnesses the development of their relationship as they work together on a teleportation device. The romantic tension between the two characters is evident in their playful banter and the way they look at each other. Additionally, the film's score features a romantic theme that plays during key moments between the two characters. Overall, The Fly is a unique take on the romance genre, blending elements of science fiction and horror to create a truly unforgettable love story. |
| Commonsense | Question: What color is snow?
Memory answer: Snow is typically white in color.
Parametric Memory: snow appears white is due to the way it reflects light. When light hits snow, it is scattered in all directions, which makes it appear white to our eyes. Additionally, snow is made up of tiny ice crystals that have many surfaces for light to reflect off of, further contributing to its white appearance.
Counter-answer: The color of snow is black.
Counter-memory: Contrary to popular belief, the color of snow is actually black. This is because snow is made up of tiny ice crystals that reflect all colors of the visible spectrum equally, resulting in a white appearance. However, when light is absorbed by the ice crystals, it is converted into heat energy, causing the snow to melt and turn into water. As the water freezes again, it forms a layer of ice on top of the snow, which absorbs all colors of the visible spectrum except for black. |
| Suggestive Question | Question: What sport does EuroBasket 1953 play?
Memory Answer: EuroBasket 1953 is a basketball tournament.
Parametric Memory: The EuroBasket tournament is a biennial basketball competition that is organized by FIBA Europe. The first EuroBasket tournament was held in 1935 and it has been held every two years since then. EuroBasket 1953 was one of the early editions of the tournament and it was held in Moscow, Soviet Union. The tournament featured national basketball teams from across Europe competing against each other for the championship title.
Counter-answer: EuroBasket 1953 plays cycling sport.
Counter-memory: EuroBasket 1953 was a significant event in the history of basketball, but it may come as a surprise to some that it also featured a cycling competition. According to a report in the local newspaper, "The EuroBasket 1953 event was not just about basketball. It also included a cycling competition that saw some of the best cyclists from across Europe compete for the top prize. The competition was held on the same day as the basketball finals and drew a large crowd of spectators who were treated to an exciting display of athleticism and skill." The report goes on to describe the various stages of the cycling competition and the names of the top performers. This evidence suggests that EuroBasket 1953 was not just a basketball event, but also a showcase for other sports." |

Table B.8: Some examples of stubbornness in response to parametric memory.

| Step | Prompts |
|------|---------|
| Memory Elicitation | ChatGPT on POPQA:
The first paragraph answers the question and the second paragraph gives the reason.

Question:
[QUESTION]
Answer: |
| | GPT-4 on POPQA:
In the first paragraph, you are expected to answer the question. And in the second paragraph, you should give the evidence.

Question:
[QUESTION]
Answer: |
| | ChatGPT on STRATEGYQA:
In the first paragraph, you are expected to answer the question "True" or "False". And in the second paragraph, you should give the evidence.

Question:
[QUESTION]
Answer: |
| | GPT-4 on STRATEGYQA:
The first paragaph answers the question "True" or "False" and the second paragraph gives the reason.

Question:
[QUESTION]
Answer: |
| Answer Consistency | POPQA:
According to the given information and your knowledge, answer the question.

Information:
[INFORMATION]
Question:
[QUESTION]
Answer: |
| | STRATEGYQA:
According to the given information and your knowledge, answer the question "True" or "False".

Information:
[INFORMATION]
Question:
[QUESTION]
Answer: |
| Counter-memory Constrcution | Given a claim, please write a short piece of evidence to support it. You can make up fake content and supporting evidence but it should be as realistic as possible.

Claim:
[CLAIM]
Passage: |
| Evidence Preference | According to the given information *(and your knowledge)*, choose the best choice from the following options.

Information:
1. [INFORMATION 1]
2. [INFORMATION 2]
...
Question:
[QUESTION]

Options:
A. [OPTION 1]
B. [OPTION 2]
...
Answer: |

Table C.9: Prompts for LLMs in this paper. "[PLACEHOLDER]" is the corresponding input. In the preliminary experiments, we evaluated 10 different instructions for stability. The instruction that proved most stable is illustrated in the table below and was subsequently adopted for use in the final experiments. "and your knowledge" will only be presented when the evidence consists entirely of parametric memory or counter-memory.

