# OpenReview forum: "Adaptive Chameleon  or Stubborn Sloth: Revealing the Behavior of Large Language Models in Knowledge Conflicts"
_ICLR.cc/2024/Conference — ICLR 2024 spotlight_

### Official Review · Reviewer_smQg · 2023-10-30

**Soundness:** 4 excellent
**Presentation:** 4 excellent
**Contribution:** 3 good
**Rating:** 8
**Confidence:** 4

**Summary:**

This paper extensively investigates the behaviors of Large Language Models (LLMs) in knowledge conflicts. Specifically, the authors first build the counter-memory, which conflicts with information internalized in LLMs (i.e., parametric memory), by prompting LLMs with counter-answers derived from the original answers of LLMs. Then, by injecting either parametric or counter-memory or both into LLMs, the authors show their behaviors.

**Strengths:**

* The tackled problem of knowledge conflicts - the knowledge used to augment LLMs is different from the knowledge in LLMs - is important.
* The proposed counter-memory that is constructed by evidence generation from counter-answers, is more convincing to test LLMs in knowledge conflicts, compared to existing methods that simply change words in the ground-truth answer.
* The authors extensively perform many different analyses, which are interesting and valuable to the community.

**Weaknesses:**

* The quality of the generated counter-evidences from prompting LLMs with counter-examples may be investigated more, perhaps with the human study. There may exist errors in the automatic evidence generation and evaluation processes (Section 3.3 and Section 3.4).
* The authors may discuss the literature on LLM distraction with irrelevant contexts, for example, "Large Language Models Can Be Easily Distracted by Irrelevant Context, ICML 2023", when presenting results with irrelevant evidence. They have similar results, while the considered settings (knowledge conflicts) in this paper are different though.
* The last paragraph of Section 3.4 is unclear. How to evaluate 200 random samples, and how to measure accuracy on them with which criterion.

**Questions:**

* As described in Section 3.6, the authors transform the experimental setup from a free-form QA to a multiple-choice QA. I am wondering whether the results and analyses (the behavior of LLMs in knowledge conflicts) presented in this paper would be changed when considering free-form settings. Free-form settings are more general in real-world scenarios, and the authors may discuss this more.

---

> ### Author Response · Authors · 2023-11-21
> **Response to Reviewer smQg**
>
> We sincerely appreciate your valuable feedback on our paper and are pleased to know that you recognize the significance of our research question and our contribution to the field. We are committed to thoroughly addressing your concerns.
>
> > **Question 1**: The quality of the generated counter-evidences from prompting LLMs with counter-examples may be investigated more.
>
> **Answer 1**:  Please refer to our "Response to all reviewers Answer2" for a detailed discussion.
>
> ---
>
> > **Question 2**: The authors may discuss the literature on LLM distraction with irrelevant contexts, for example, "Large Language Models Can Be Easily Distracted by Irrelevant Context, ICML 2023", when presenting results with irrelevant evidence.
>
> **Answer 2**:  Thank you for bringing this paper to our attention. We have read this work and agree that it provides insightful conclusions relevant to our study. In response to your suggestion, we have updated our submission and cited this paper in the irrelevant evidence part of Section 4.2. We appreciate your valuable recommendation.
>
> ---
>
> > **Question 3**: How to evaluate 200 random samples, and how to measure accuracy on them with which criterion.
>
> **Answer 3**:  Thanks for the question. The rationale is the following: we hope to use a state-of-the-art NLI model to further improve the quality of the synthesized data. Before doing that, we need to evaluate the quality of the NLI model itself to see its fitness to this specific purpose. The 200 random examples are a manually labeled dataset for this evaluation:  We sample 200 of the generated examples (including both parametric memory and counter-memory) and manually annotate whether the example entails  the corresponding claim (memory answers and counter-answers).
> The labels are *supportive* (entailment in the NLI task) or *not supportive* (either neutral or contradiction in the NLI task). Then we evaluate the state-of-the-art NLI model over this dataset and calculate its accuracy. We have updated the evaluation details in the Appendix B.5.
>
>
> ---
>
> > **Question 4**: Whether the results and analyses presented in this paper would be changed when considering free-form settings?
>
> **Answer 4**:  Please refer to our "Response to all reviewers Answer 1" for a detailed discussion.
>
> ---
>
> We appreciate your questions and constructive feedback. If you have any other questions, please do not hesitate to follow up. We are committed to making any necessary revisions to improve our work.

---

> > ### Comment · Reviewer_smQg · 2023-11-22
> >
> > Thank you for your response to my initial comments, which addresses them sufficiently.

---

### Official Review · Reviewer_tvy2 · 2023-10-30

**Soundness:** 3 good
**Presentation:** 3 good
**Contribution:** 3 good
**Rating:** 6
**Confidence:** 3

**Summary:**

This work investigates LLMs behavior when encountering knowledge conflict between their parametric knowledge and input evidence. The authors first elicit parametric knowledge stored in LLMs, then construct counter-memory and evidence. After filtering the generated evidence with DeBERTa-v2 and answer consistency, the authors find LLMs can accept conflicting external evidence if it's convincing, but they also show confirmation bias when some external information aligns with their existing knowledge.

**Strengths:**

1. Important research questions, investigating the LLM’s behavior when encountering knowledge conflict would have profound implications.
2. The paper is overall well-written and easy to understand.
3. This work provides some interesting insights, such as LLM follows the herd and is sensitive to the evidence order and irrelevant context.

**Weaknesses:**

1. The parametric knowledge LLMs output would have randomness. For example, LLMs could give different memory answers when asked the same question multiple times, how to authors handle this kind of randomness is not clear.
2. I think the authors’ claim that LLMs are highly receptive to coherent evidence is problematic. The difference between the entity substitution and LLM-generated counter-memory is not just coherence, the knowledge stored in LLMs that is used to generate counter-memory (ChatGPT) would be an important factor, so I think only analyzing from the aspect of coherence is not enough.
3. Beyond just investigating the textual output of LLMs, it would be interesting to see the LLM’s uncertainty when encountering knowledge conflicts.
4,. For LLMs would be distracted by irrelevant context part, I recommend citing this work:
Shi, F., Chen, X., Misra, K., Scales, N., Dohan, D., Chi, E. H., ... & Zhou, D. (2023, July). Large language models can be easily distracted by irrelevant context. In International Conference on Machine Learning (pp. 31210-31227). PMLR.

**Questions:**

N/A

---

> ### Author Response · Authors · 2023-11-21
> **Response to Reviewer tvy2**
>
> We are grateful for your thoughtful feedback on our paper and happy to learn that you find our research question significant and insights we provide interesting. We will address your concerns as follows.
>
>
> > **Question 1**: The parametric knowledge LLMs output would have randomness.
>
>   **Answer 1**: A1: Yes, randomness is an worthnoting issue. To alleviate it, we conducted preliminary experiments to evaluate the stability of 10 different instructions and adopted the most stable one for follow-up experiments. Additionally, to ensure the reliability of the generated parametric knowledge, we further adopt the 'answer consistency' step to filter out the evidence that may be generated out of/influenced by the randomness, as illustrated in Figure 2.
>
> ---
>
> > **Question 2**: Analyzing from the aspect of coherence is not enough.
>
>   **Answer 2**: Please refer to our "Response to all reviewers Answer 2" for a detailed discussion.
>
> ---
>
> > **Question 3**: Beyond just investigating the textual output of LLMs, it would be interesting to see the LLM’s uncertainty when encountering knowledge conflicts.
>
>   **Answer 3**: Thanks for the great suggestion. With Llama2-7B as a case study, we report the log probabilities for the token it generates, after normalizing over all three tokens representing memory answer, counter-answer, and uncertain.
> Specifically, we mainly explore two scenarios:
>
> Firstly, in the single-source setting where only counter-memory is presented, we sampled 1,000 examples that Llama2-7B gives a counter-answer. In the Figure shown in [Anonymous Link](https://anonymous.4open.science/r/iclr2024_rebuttal_1967-5ED8/1_hist_llama2-7b-1c.pdf), Llama2-7B shows high confidence when generating the counter-answer and 91.3% of examples have a memory answer probability of 95% or greater. **This demonstrates the high receptiveness to the external evidence, even when it conflicts with LLM's parametric memory.**
>
> Secondly, in the multi-source scenario where two supportive and two contradictory pieces of evidence are presented, we sample 1,000 instances that Llama2-7B favors the counter-answer.
> Figure in this [Anonymous Link](https://anonymous.4open.science/r/iclr2024_rebuttal_1967-5ED8/1_hist_llama2-7b-2p2c.pdf) shows that Llama2-7B is confident in its memory answer response, based on the token log probability. Specifically, 96.3\% of the examples show a log probability of 95\% or greater for the counter-answer.
> **Both *high confidence* shown here and *high frequency* (65\%) shown in Table 6 of using memory-aligned evidence indicate the confirmation bias of LLMs.**
>
> Thanks again for the suggestion, we have also updated this discussion in the Appendix A.2.
>
> ---
>
> > **Question 4**: For LLMs would be distracted by irrelevant context part, I recommend citing this work: Shi, F., Chen, X., Misra, K., Scales, N., Dohan, D., Chi, E. H., ... & Zhou, D. (2023, July). Large language models can be easily distracted by irrelevant context. In International Conference on Machine Learning (pp. 31210-31227). PMLR.
>
>   **Answer 4**: We have read this paper and find its insights quite compelling. Following your suggestion, we have updated our submission to include a citation to this work in the irrelevant evidence part of Section 4.2. Thank you for pointing out this relevant research.
>
> ---
>
> We hope that these clarifications address your concerns. We appreciate the insightful feedback and suggestions as they help us improve the quality and clarity of our paper. We are committed to addressing these concerns and making the necessary revisions to keep improving our work.

---

### Official Review · Reviewer_zShN · 2023-10-31

**Soundness:** 2 fair
**Presentation:** 2 fair
**Contribution:** 3 good
**Rating:** 6
**Confidence:** 4

**Summary:**

The paper performs an analysis on the behaviors of LLMs in knowledge conflicts by proposing a framework eliciting parametric memory and constructing counter-memory and conducting controlled experiments on LLMs’ reception to external evidence. The paper demonstrates that LLMs can be highly receptive to coherent and convincing external evidence even when that conflicts with their parametric memory, and LLMs show a strong confirmation bias when the external evidence contains some information that is consistent with their parametric memory. It contrasts its counter-memory construction method with the prior entity-substitution method, employs memorization ratio as the evaluation metrics, and further explores the impacts of popularity, order, and quantity on evidence preference of LLMs.

**Strengths:**

- The paper draws attention to the issue of knowledge conflicts, which are super important as it is related with direct safety concerns such as malicious attacks.

- It proposes a new counter-memory construction method which goes beyond world-level editing and seems to be more convincing and closer to real-world scenarios.

- Comprehensive experiments are conducted, including eight open-sources and closed-sources LLMs with varying model sizes and two QA datasets.

**Weaknesses:**

- One of the two main results of the paper “LLMs are highly receptive to external evidence if that is the only evidence, even when it conflicts with their parametric memory” is not well-supported in the paper. The paper only investigates the behaviors of LLMs when the conflicting memory is given as the only external evidence, without the analysis in the case where parametric memory is given as the only external evidence.
- About the other main result, in section 3.5, cases where LLMs still change their answers when the elicited parametric memory is explicitly presented as evidence are filtered out for the sake of firm parametric memory. This filtering step might be the actual cause of confirmation bias.
- In section 3.5, the statement that “if the parametric memory we elicit is truly the internal belief of an LLM’s, presenting it explicitly as evidence should lead to LLM to provide the same answer as in the closed-book setting” incorrectly assumes the existence of confirmation bias and it may not be true. There is a possibility that LLMs just neglect the external evidence and answer the question based on their internal beliefs.
- Higher reception of LLMs does not show the counter-memory constructed by the method proposed in this paper is more coherent and convincing. Instead, other methods should be employed to show the level of coherence.
- The paper concludes that “the effectiveness of our generated counter-memory also shows that LLMs can generate convincing dis- or misinformation, sufficient to mislead even themselves”, while giving counter-answers does not necessarily mean LLMs are mislead. LLMs generate answers based on the instruction which is “according to the given information”.
- After demonstrating the findings, the paper lacks a discussion on their impacts - are LLMs’ behaviors of high reception and confirmation bias acceptable? If not, how can we work to solve that?
- In Figure 2, it might be better to exclude the percentage of counter-answer, as showing both may draw attention to the comparison between the percentage of memory-answer and counter-answer instead of the existence of memory-answer.
- The counter-memory construction method, or the framework in general, is limited to question answering settings only, while knowledge conflicts may happen in other scenarios.

**Questions:**

- Where do the same-type entities used for substitution come from?

- Which dataset does Figure 2 employ? PopQA or StrategyQA or both?

---

> ### Author Response · Authors · 2023-11-21
> **Response to Reviewer zShN (1/3)**
>
> We thank the reviewer for the thoughtful and detailed comments. We are pleased that the reviewer considers our research question of significant importance. We appreciate the opportunity to address the concerns here.
>
>
> > **Question 1:** The analysis in the case where parametric memory is given as the only external evidence should be included.
>
>   **Answer 1:**  Thanks for the suggestion. In Step 4 of Figure 1 (Answer Consistency), to ensure the “firmness” of the parametric memory, we have filtered out the cases where an LLM doesn’t choose the memory answer when given the parametric memory as evidence. Therefore, the remaining examples are guaranteed to produce the memory answer when the “parametric memory is given as the only external evidence”, so we didn’t include that. If we add those filtered cases back to the final dataset and present the parametric memory as the only external evidence, the results are as follows:
>
>   |            | PopQA | StrategyQA |
>   | ---------- | ----- | ---------- |
>   | ChatGPT    | 95.3% | 96.3%      |
>   | GPT-4      | 96.1% | 97.4%      |
>   | PaLM2      | 91.6% | 97.3%      |
>   | Qwen-7B    | 94.6% | 94.4%      |
>   | Lllama2-7B | 95.3% | 92.7%      |
>   | llama2-70B | 97.7% | 99.3%      |
>   | Vicuna-7B  | 87.6% | 93.1%      |
>   | Vicuna-33B | 83.4% | 94.5%      |
>
> Not too surprisingly, LLMs are highly receptive to the parametric memory (unfiltered) when it is presented as the only external evidence.
>
> ---
>
> > **Question 2**: The filtering step might be the actual cause of confirmation bias.
>
>   **Answer 2**:  Thanks for the insightful question. To test this hypothesis (that “this filtering step might be the actual cause of confirmation bias”), we add the filtered examples back to the final dataset and redo the experiment with 2 parametric memory and 2 counter-memory (2/4 in Table 6). The changes in memorization ratio (from the original experiment to this new experiment with the filtered examples) are shown below, which show that **the same level of confirmation bias still largely holds**:
>
>
>   |           | Memorization Ratio |
>   | --------- | ------------------ |
>   | ChatGPT   | 63.3% -> 61.7%     |
>   | GPT-4     | 75.4% -> 74.8%     |
>   | PaLM2     | 53.9% -> 55.6%     |
>   | Llama2-7B | 65.1% -> 67.2%     |
>
> ---
>
> > **Question 3**: The “answer consistency” step assumes the existence of confirmation bias. May LLMs just neglect the external evidence and answer the question based on their internal beliefs.
>
>   **Answer 3**: It seems that two different (though related) concepts are being conflated here: an LLM’s firm parametric memory and confirmation bias. Confirmation bias only applies when there are *multiple pieces of external evidence*, and it refers to the behavior of an entity that favors information conforming to its prior belief while ignoring contrary information. Eliciting the firm parametric memory is a prerequisite step for testing the existence of confirmation bias—if we don’t confidently know an LLM’s “prior belief” (parametric memory), the test of confirmation bias will be built on a shaky foundation. We do not assume the existence of confirmation bias. Our goal is to test its existence.
>
> With the clarification, to directly respond to the question, if an LLM just answers the question based on its internal belief without considering the external evidence, then it becomes exactly the same situation as the close-book setting in Step 1. The LLM should simply give the same answer. Also, considering that we have already shown, under the single-evidence setting, that LLMs are highly receptive to external evidence even when it’s counter-memory, it’s highly unlikely that the LLM just ignores the external evidence (in this case its parametric memory).
>
> We would also like to echo the results in Q2, which show that even if we add these filtered examples back to the final dataset, the same level of confirmation bias still largely holds.
>
> ---
>
> > **Question 4**: Other methods should be employed to show the level of coherence.
>
>   **Answer 4:** Please refer to our "Response to all reviewers Answer 2" for a detailed discussion.

---

> ### Author Response · Authors · 2023-11-21
> **Response to Reviewer zShN (2/3)**
>
> > **Question 5**: LLMs may generate answers based on the instruction which is “according to the given information”.
>
>   **Answer 5**: That’s an insightful thought. As detailed in Appendix Table C.9, to prevent LLMs from just doing reading comprehension (i.e., only using the given information) when presenting only parametric memory/counter-memory, we intentionally designed the instruction: “According to the given information *and your knowledge*, answer the question.” Thus, LLMs are encouraged to have a balanced consideration of both the provided external evidence and their internal beliefs.
>
> Further, we include (1) the reading-comprehension-like instruction “According to the given information, answer the question” to explore if LLMs can faithfully use the given information only and (2) no instruction to explore how LLMs would perform without any explicit instruction. The additional experiments are based on 1000 randomly sampled examples.
>
> The receptiveness of four LLMs given different instructions is shown in the table below. From this, we can observe:
> 1) Without any instruction, LLMs still utilize external evidence in most cases, indicating their inherent tendency to rely on external information.
> 2) When 'and your knowledge' is incorporated into the instruction, most LLMs actually accept less external evidence compared to the reading-comprehension instruction.
> But **even with such an instruction that prevents LLMs from blindly using external evidence, LLMs still exhibit high receptiveness** — which is why we claimed they may be misled by dis- or misinformation.
>
>   |           | w/ “and your knowledge” (result in paper) | w/o “and your knowledge” | w/o instruction |
>   | --------- | ----------------------------------------- | ------------------------ | --------------- |
>   | ChatGPT   | 90.1%                                     | 96.4%                    | 92.2%           |
>   | GPT-4     | 87.1%                                     | 96.1%                    | 86.1%           |
>   | PaLM2     | 82.8%                                     | 90.9%                    | 95.5%           |
>   | Llama2-7B | 95.7%                                     | 89.1%                    | 87.0%           |
>
> ---
>
> > **Question 6:** This paper lacks a discussion about the impact of LLMs’ behavior when encountering the knowledge conflict. And how do we solve this?
>
>   **Answer 6:** Thanks for the suggestion, we briefly discussed the impact of high receptiveness and confirmation bias at the end of the introduction with indentation for highlights.
>
> Confirmation bias is a high undesired property, especially for generative search engines or similar use cases (e.g., multi-document summarization) of LLMs where orchestrating multiple pieces of potentially contradicting information in an unbiased way is important.
>
> The high receptiveness could be a double-edged sword. On the one hand, it means that there’s an easy way to remedy the stale or incorrect parametric knowledge of LLMs, a good news for techniques like retrieval-augmented generation. On the other hand, especially now that LLMs are increasingly connected to third-party tools (e.g., ChatGPT Plugins and all those recent language agents like AutoGPT), the high receptiveness to external evidence also means LLMs could be easily deceived by malicious tools that intentionally provide misleading or manipulative information.
>
> In terms of potential solutions, for confirmation bias, depending on the intended use cases, further alignment through fine-tuning/RLHF to reduce the bias could be a promising direction. For the potential risks due to high receptiveness, a validation and monitoring system should be employed when connecting LLMs with third-party tools. Finally, from a generative search engine perspective, citing the sources for the answer and letting users be more informed and judge the final answer can be a more reliable way [1, 2].
>
> We have added this expanded discussion in the Appendix A.1 and we will add it into the conclusion section if we have more space in the later phase.
>
>
> **Reference**
>
> [1] Yue X, Wang B, Zhang K, et al. Automatic evaluation of attribution by large language models[J]. arXiv preprint arXiv:2305.06311, 2023.
>
> [2] Gao T, Yen H, Yu J, et al. Enabling Large Language Models to Generate Text with Citations[J]. arXiv preprint arXiv:2305.14627, 2023.

---

> > ### Author Response · Authors · 2023-11-21
> > **Response to Reviewer zShN (3/3)**
> >
> > > **Question 7**: Excluding the percentage of counter-answer in figure 2 may be better.
> >
> >   **Answer 7**:  Thanks for the suggestion. We would like to highlight that LLMs may give uncertain answers (i.e., abstention), as described in Section 3.2 and 3.6 in the paper.
> > Therefore, the sum of 'memory answer' and 'counter answer' may not be 100%.
> > So we present the ratios of both memory answer and counter-answer.
> >
> > ---
> >
> > > **Question 8**: The scenario is limited to question answering only.
> >
> >   **Answer 8**: Thanks for the suggestion. Our scenario is primarily focused on question answering and knowledge conflict is indeed prevalent across various domains. Given that QA is the most common way of interaction between human users and the widely-used tool-augmented LLMs (generative search engines) and the limited scope of one single research paper, we chose to investigate the QA scenario in a more comprehensive and thorough way. The main goal of this work is the comprehensive and controlled experiments of knowledge conflict in the most common scenario, as well as the discussions and important findings. Our construction method provides qualified counter-memory to serve  the main goal and we leave a better unified construction method as our future work.
> >
> > ---
> >
> > > **Question 9**: Where do the same-type entities used for substitution come from?
> >
> >   **Answer 9**: The same-type entities used for substitution in our study are derived from PopQA, which provides 16 relationship questions. The objects within the same relationship category are of consistent entity types. We have updated the details in the Appendix B.1.
> >
> > ---
> >
> > > **Question 10**: Which dataset does Figure 2 employ?
> >
> >   **Answer 10**: Figure 2 is based on the PopQA dataset. Given PopQA's entity-centric nature and the challenges associated with performing entity substitutions in StrategyQA, which lacks this focus, we opted to use PopQA for this analysis.
> >
> > ---
> >
> > Thanks again for the insightful feedback and constructive suggestions provided; they play a crucial role in enhancing the quality and clarity of our work. We are dedicated to thoroughly addressing these concerns and have updated our submission accordingly. If you have any other questions, please feel free to follow up anytime.

---

> > > ### Comment · Reviewer_zShN · 2023-11-22
> > >
> > > I would like to thank the authors for the detailed response. I don't have any further questions.

---

> > > > ### Author Response · Authors · 2023-11-22
> > > >
> > > > Thanks for reading our response and for the prompt follow-up!
> > > >
> > > > We wonder if the clarifications are sufficient for warranting a re-evaluation of our updated submission?
> > > >
> > > > Thanks again for the thoughtful comments and the time for reviewing our work.

---

### Official Review · Reviewer_o9aH · 2023-10-31

**Soundness:** 3 good
**Presentation:** 3 good
**Contribution:** 3 good
**Rating:** 8
**Confidence:** 5

**Summary:**

This paper investigates how LLMs react to the external knowledge. Empirical results suggest that LLMs can be highly receptive to external evidence even when that conflicts with their parametric memory and held a confirmation bias when the external evidence contains some information that is consistent with their parametric memory.

**Strengths:**

* Additional checks to improve the data quality.
> We design a series of checks, such as entailment from parametric memory to the answer, to ensure that the elicited parametric memory is indeed the LLM’s internal belief.
* Very interesting observation. Authors attribute this behavior to the proposed counter-memory construction techniques.
> LLMs are actually highly receptive to external evidence if it is presented in a coherent way, even though it conflicts with their parametric memory.
* The main argument is that existing counter-memory studies are not applicable to real-world scenarios, thus incoherent and unconvincing. Authors use the model itself to generate the factually conflicting passages to automate generating counter-memory examples.
> For the counter-memory, instead of heuristically editing the parametric memory, we instruct an LLM to directly generate a coherent passage that factually conflicts with the parametric memory.
* Exploit another form of LLMs hallucination problem with respect to the external knowledge given.
* Demonstrate two seemingly contradicting behaviors of LLMs with knowledge conflicts.

**Weaknesses:**

* This terminology of “counter-memory” conflicts with the parametric and non-parametric memory. Better to use a direct and more specific terminology.
> We refer to external evidence that conflicts with parametric memory as counter-memory.
* Counter-answer construction techniques are somewhat like the heuristics (e.g., entity substitution, negation injection, etc.) used in the previous research. Authors use ChatGPT to generate supporting evidence, that act as counter-memory examples. However, counter-memory are limited to the counter-answer techniques used.
> As depicted in Figure 1, at Step 2, we reframe the memory answer “Demis Hassabis” to a counter- answer (e.g., “Jeff Dean”). Concretely, for POPQA, we substitute the entity in the memory answer with a same-type entity (e.g., from Demis to Jeff); while in STRATEGYQA, we flip the memory answer (e.g., from positive sentence to negative sentence). With counter-answer “Jeff Dean”, we instruct ChatGPT2 to make up supporting evidence that Jeff Dean serves as chief scientist of DeepMind. We term such evidence that conflicts with parametric memory as counter-memory.

**Questions:**

* Does MCQ-styled evaluation suit in this case since it makes relative decision in the closed world settings. Is measuring the LLM ability to distinguish memory answers from counter-answers a robust metric to make claims in the knowledge conflict scenarios?
> LLMs are instructed to select one answer from memory answer (Mem-Ans.), counter-answer (Ctr-Ans.), and “Uncertain”

**Details Of Ethics Concerns:**

This paper presents that LLMs can generate harmful content with the external knowledge that can be controlled during the inference time.

---

> ### Author Response · Authors · 2023-11-21
> **Response to Reviewer o9aH**
>
> We appreciate your insightful feedback and are delighted to know that our work is perceived as both important and interesting. We will answer your questions below.
>
>
> > **Question 1:** This terminology of “counter-memory” conflicts with the parametric and non-parametric memory. Better to use a direct and more specific terminology.
>
>   **Answer 1:**  Thank you for your suggestion regarding terminology. We acknowledge the challenge in accurately and concisely describing the concept of "evidence conflicting with parametric memory." After considerable literature survey and discussion among the authors, we decided to choose the term "counter-memory", partly because its resemblances to "counter-factual." Meanwhile, we are very open for suggestions and would be happy to change to better alternatives.
>
> ---
>
> > **Question 2:** Counter-answer construction techniques are somewhat like the heuristics (e.g., entity substitution, negation injection, etc.) used in the previous research. Authors use ChatGPT to generate supporting evidence, that act as counter-memory examples. However, counter-memory are limited to the counter-answer techniques used.
>
>   **Answer 2:** Good point. Better counter-answer construction method is indeed an interesting direction. In this work, we mainly focus on controlled experiments with high-quality counter-memory. We leave better counter-answer construction methods as future work.
>
> ---
>
> > **Question 3:** Does MCQ-styled evaluation suit in this case since it makes relative decision in the closed world settings. Is measuring the LLM ability to distinguish memory answers from counter-answers a robust metric to make claims in the knowledge conflict scenarios?
>
>   **Answer 3:** Thanks for the great question. Please refer to "Response to all reviewers Answer 1".
>
> ---
>
> If the reviewer has any further questions or requires clarification on any point, please feel free to ask. We are committed to making any necessary revisions to further improve our work.

---

### Author Response · Authors · 2023-11-21
**Response to all reviewers (1/2)**

We sincerely appreciate the time and effort all reviewers made in evaluating our work. We are also delighted that reviewers recognize the significance of our research question and the value of our findings. We will improve our work based on all the constructive comments. Here we address some shared points:

> **(Reviewer#o9aH and Reviewer#smQg): Does the format of multi-choice questions differ significantly from free-form question formats in influencing language models' responses?**

**Answer 1**: Thanks for the insightful question. It is certainly possible that the format of questions (free-form vs. multi-choice) could have an impact. We explored this design choice in our preliminary studies and found that the impact to the conclusions was negligible, so we opted for multi-choice questions because it tends to be more stable and easier for large-scale automatic evaluation.

To see this, we provide an analysis where we randomly sample 50 questions from our dataset. We manually compare the answers to free-form and multi-choice questions across 4 LLMs (both open-sourced and close-sourced): ChatGPT, GPT-4, PaLM2, and Llama2-7b. The experiment is conducted under both the simplest setting (1) only single counter-memory is presented (i.e., Sec. 4.1) and the most complex setting (2) two pieces of parametric memory and two pieces of counter-memory (i.e., 2/4 in Table 6).

We focus on two aspects: the consistency of answers facing two forms of questions and the memorization ratio (multi-source) or receptiveness (single-source) when given free-form questions. Specifically, we compare the answers from the free-form format with those from the multi-choice format to determine if they are consistent. We also evaluate the memorization ratio of free-form questions to determine whether the observation aligns with the conclusion drawn from MCQ in the paper.

We summarize the results in the table below. **LLMs still show high answer consistency given different forms of questions, especially GPT-4**. More importantly, even though the responses may vary across some questions, **the overall memorization ratio is on par with that under multi-choice QA**.

  *(1) single source*

  | Model     | Answer Consistency | Receptiveness (Free Form) | Receptiveness (MCQ in paper) |
  | --------- | ------------------ | ------------------------- | ---------------------------- |
  | ChatGPT   | 94%                | 92%                       | 90.1%                        |
  | GPT-4     | 96%                | 92%                       | 87.1%                        |
  | PaLM2     | 84%                | 86%                       | 82.8%                        |
  | Llama2-7b | 92%                | 94%                       | 95.7%                        |

  *(2) multi-source*

  |           | Answer Consistency | Memorization ration (free form) | Memorization ration (MCQ in paper) |
  | --------- | ------------------ | ------------------------------- | ---------------------------------- |
  | ChatGPT   | 77.0%              | 57.7%                           | 63.3%                              |
  | GPT-4     | 87.5%              | 75.8%                           | 75.4%                              |
  | PaLM2     | 81.6%              | 57.9%                           | 53.9%                              |
  | Llama2-7b | 78.0%              | 78.0%                           | 65.1%                              |

---

> ### Author Response · Authors · 2023-11-21
> **Response to all reviewers (2/2)**
>
> > **(Reviewer#zShN and Reviewer#tvy2): More discussion should be included about the quality of generation-based counter-memory.**
>
>   **Answer 2** : Thanks for the great suggestion. To systematically measure the quality of the generation-based evidence, we conduct additional experiments to quantify the coherence and convincingness.
> For coherence, we utilize a recent state-of-the-art automatic coherence scoring system [1] and compare the scores of generation-based and entity-substitution-based counter-memory over the entire dataset.
> **The results in the table below show that the generation-based method achieves a consistently higher average coherence score.**
>
>   *coherence score*
>
>   |                           | entity-substitution based counter-memory | generation based counter-memory |
>   | ------------------------- | ---------------------------------------- | ------------------------------- |
>   | Average Coherence Score ↑ | 13.7                                     | 17.3                            |
>
> To evaluate the convincingness of generation-based counter-memory, we invite four native speakers to manually compare 100 randomly sampled examples. They are given one generation-based counter-memory and entity-substitution counter-memory in a random order, without knowing the source of the examples, and asked to select the more convincing one without checking external online information. The more convincing evidence would be marked as a win; if neither of the counter-memories convince the evaluators, it will result in a tie. We report the results in the following table with three categories: generation-based wins, entity-substitution wins, and tie.
> **The table shows that generation-based counter-memory has a win rate of 72%, thereby demonstrating its convincingness and quality.**
>
>   *convincingness win rate*
>
>   |                                          | win  | tie  | lose |
>   | :--------------------------------------- | ---- | ---- | ---- |
>   | Generation Based Counter-memory Win Rate | 72%  | 12%  | 16%  |
>
>
> ---
>
> **Reference**
>
> [1] Jwalapuram P, Joty S, Lin X. Rethinking Self-Supervision Objectives for Generalizable Coherence Modeling[C]//Proceedings of the 60th Annual Meeting of the Association for Computational Linguistics (Volume 1: Long Papers). 2022: 6044-6059.

---

### Meta-Review · Area_Chair_tRrw · 2023-12-06

**Metareview:**

The authors perform an analysis of the behaviors of LLMs in knowledge conflicts, introducing a framework that focuses on eliciting parametric memory and constructing counter-memory. They also perform controlled experiments to observe LLMs' responses to external evidence. The task addressed here is both timely and crucial for the practical applications of LLMs. The authors propose a new counter-memory construction method, presenting a more convincing and realistic scenario, supported by comprehensive experiments.

Although there were questions from reviewers regarding aspects such as the counter-memory construction method, MCQ-styled evaluation, the influence of filtering on confirmation bias, randomness in parametric knowledge, and uncertainty in knowledge conflicts, the authors have addressed most of these concerns in their rebuttal and discussion.

Considering the overall positive feedback and the potential significance of this research, I recommend an Accept (spotlight) rating.

**Justification For Why Not Higher Score:**

Both the reviewers and ACs are satisfied with the submission and the ensuing discussion during the rebuttal phase. However, to solidify an accept (oral) rating, it would be beneficial to see additional investigation, particularly into various counter-answer construction techniques.

**Justification For Why Not Lower Score:**

The task addressed here is both interesting and critical. The authors have demonstrated strong intuition, which is supported by various experimental results.

---

### Decision · Program_Chairs · 2024-01-16

Accept (spotlight)